# Unveiling Differences: A Vision Encoder-Decoder Model for Difference Medical Visual Question Answering

**Luis-Jesus Marhuenda**[*1] iD                                    LJMARTEN@PRHLT.UPV.ES
**Miquel Obrador-Reina**[*1] iD                                   MOBRREI@PRHLT.UPV.ES
**Mohamed Aas-Alas**[*1] iD                                      MAASALA@PRHLT.UPV.ES
**Alberto Albiol**[1] iD                                        ALALBIOL@PRHLT.UPV.ES
**Roberto Paredes**[1] iD                                       RPAREDES@PRHLT.UPV.ES
[1] *Campus de Vera, Universitat Politècnica València, Camí de Vera s/n, 46022 Valencia, Spain*

**Editors:** Accepted for publication at MIDL 2025

## Abstract

Difference Medical Visual Question Answering (Diff-VQA), a specialized subfield of Medical VQA, tackles the critical task of identifying and describing differences between pairs of medical images. This study introduces a novel Vision Encoder-Decoder (VED) architecture tailored for this task, focusing on the comparison of chest X-ray images to detect and explain changes. The proposed model incorporates two key innovations: (1) a light-weight Transformer text decoder architecture capable of generating precise and contextually relevant answers to complex medical questions, and (2) an enhanced fusion mechanism that improves the model's ability to distinguish between two input images, enabling more accurate comparison of radiological findings. Our approach excels in identifying significant changes, such as pneumonia and lung opacity, demonstrating its utility in automating preliminary radiological assessments. By leveraging large-scale, domain-specific datasets and employing advanced training strategies, our VED architecture achieves state-of-the-art performance on standard VQA metrics, setting a new benchmark in diagnostic accuracy. These advancements highlight the potential of Diff-VQA to enhance clinical workflows and support radiologists in making more precise, informed decisions.

**Keywords:** Difference Visual Question Answering, Vision Encoder-Decoder Model, Transformers, Medical Imaging

## 1. Introduction

Visual Question Answering (VQA) is a challenging task that bridges the domains of computer vision (CV) and natural language processing (NLP). It involves the generation of accurate and contextually relevant answers to specific questions posed about visual data. Unlike general vision tasks that focus solely on image analysis, VQA requires a deeper understanding of both visual content and linguistic semantics, enabling a meaningful interaction between visual cues and textual inputs. This complexity makes VQA an essential tool for real-world applications that demand precise, task-specific insights.

Building upon the challenges of Medical VQA, Difference Medical Visual Question Answering introduces an additional layer of complexity by focusing on questions that require identifying and describing differences between pairs of medical images. This task extends

---

[*] Contributed equally

the interpretative demands of Medical VQA by incorporating a comparative dimension, where systems must not only analyze individual images but also discern and articulate clinically relevant changes between them. Addressing this problem requires a nuanced understanding of temporal and spatial variations, integration of contextual information across imaging pairs, and the ability to generate precise, actionable descriptions of the differences. As such, Difference Medical VQA represents a significant advancement in the pursuit of automated, clinically meaningful insights.

To address these challenges, we present a model based on a Vision Encoder-Decoder (VED) architecture specifically designed for the Medical VQA task. This model is trained in three stages to achieve superior performance. In the first stage, the vision encoder is fine-tuned on a large-scale medical imaging dataset to capture domain-specific visual features essential for accurate medical reasoning. In the second stage, the fine-tuned vision encoder is freezed and integrated with a text decoder and trained on a specialized Medical VQA dataset. Then, in the third stage, the encoder is unfreezed and the entire model is fine-tuned to optimize the fusion of visual and textual information for generating precise answers. This comprehensive training process enables the model to produce clinically precise and contextually relevant answers, ensuring robustness and adaptability to the real-world demands of medical QA.

In this paper, the key contributions of this study are a light-weight Transformer text decoder architecture capable of generating precise and contextually relevant answers to complex medical questions and a mechanism to enhance the model's ability to distinguish between two input images during the fusion process.

To foster transparency and collaboration, we are open-sourcing our methodology on GitHub[1]. We aim to accelerate advancements in diagnostic precision and radiological interpretation, empowering the medical imaging community to enhance patient care through innovative AI-driven solutions.

## 2. Related Work

The Expert Knowledge-Aware Graph Representation (EKAID) (Hu et al., 2023) model, introduced alongside the Medical-Diff-VQA dataset, is designed for difference VQA tasks in medical imaging. It represents anatomical structures as graph nodes and constructs multi-relationship graphs to capture spatial, semantic, and implicit relationships. Features are extracted from anatomical and disease regions using Faster-RCNNs (Ren et al., 2016) trained on medical imaging datasets and are further refined through a Relation-Aware Graph Attention Network (ReGAT) (Li et al., 2019), with an LSTM (Hochreiter and Schmidhuber, 1997) and attention modules serving as the decoder for answer generation. By incorporating medical knowledge graphs, the model leverages domain-specific insights to enhance interpretability and provide accurate answers about image differences. EKAID demonstrates its capability in handling subtle disease progressions and mitigating variations caused by pose or orientation differences, outperforming MCCFormers (Qiu et al., 2021) and IDCPCL citeidcpcl in a baseline comparison.

Building upon difference-aware medical imaging models, an Expert Insight-Enhanced (EIE) (Wang et al., 2024) framework has been proposed for follow-up chest X-ray sum-

---

1. https://github.com/ljmtendero/A-VED-Model-For-Difference-Medical-VQA

mary generation. This model integrates an expert-guided difference capture module, a two-layer transformer, and a cross-modality follow-up summary generator, which employs a three-layer transformer to improve result coherence. Similarly, RegioMix (Yung et al., 2024) introduces a retrieval-augmented approach that employs mix-and-match strategies to generate pseudo-difference descriptions. It utilizes region-specific retrieval augmentation and a dual alignment module to align retrieved descriptions with input image pairs and questions, employing an encoder backbone adapted from EKAID and a decoder comprising two LSTM modules.

Further advancing temporal medical VQA, PLURAL (Cho et al., 2024) employs a Transformer-based encoder-decoder framework trained in three stages: (1) pretraining on general-purpose datasets (e.g., COCO (Lin et al., 2015), CC12M (Changpinyo et al., 2021)) to establish foundational vision-language understanding, (2) pretraining on longitudinal chest X-ray data (MIMIC-CXR, MIMIC-Diff-VQA) with additional input branches for temporal information and insights from radiology reports, and (3) fine-tuning with DiffVQA-specific data. The model processes pairs of past and current images using ResNet-101 (He et al., 2015) encoders while integrating positional and temporal encodings, with outputs passed to OFA (Wang et al., 2022), a Transformer encoder-decoder for answer generation.

## 3. Our Approach To Medical Visual Question Answering

Our proposed model for the Difference Medical VQA task utilizes a VED architecture as shown in Figure 1. It integrates the Swin Transformer (Liu et al., 2021) as a vision model and a transformer decoder as the language model.

### 3.1. Vision Component

We evaluated multiple vision encoders, including EfficientNet (Tan and Le, 2020), ViT (Dosovitskiy et al., 2021), and TinyViT (Wu et al., 2022), as potential candidates for our vision module. Preliminary experiments with EfficientNet resulted in significantly degraded performance. Similarly, TinyViT and standard ViT architectures required substantially larger models to achieve comparable accuracy. Given these limitations, we selected the Swin Transformer due to its strong performance across various vision tasks, including image classification, object detection, and segmentation, while maintaining an excellent trade-off between accuracy and computational efficiency.

Swin's hierarchical architecture and its efficient shifted window-based self-attention mechanism make it particularly well-suited for X-ray image analysis. Our implementation employs the Swin Base variant (SwinB-384), pretrained on ImageNet-21K (Ridnik et al., 2021) and fine-tuned on the MIMIC-CXR dataset using CheXpert (Irvin et al., 2019) labels. Input images are processed as three-channel $384 \times 384$ pixel images. The Swin model consists of four stages with depths of 2, 2, 18, and 2, a patch size of $4 \times 4$, and a window size of $12 \times 12$.

### 3.2. Differentation Component

In addition, we introduce an Image Differentiation Embedding (IDE) mechanism to help the model distinguish between the two input images during the fusion process. During training,

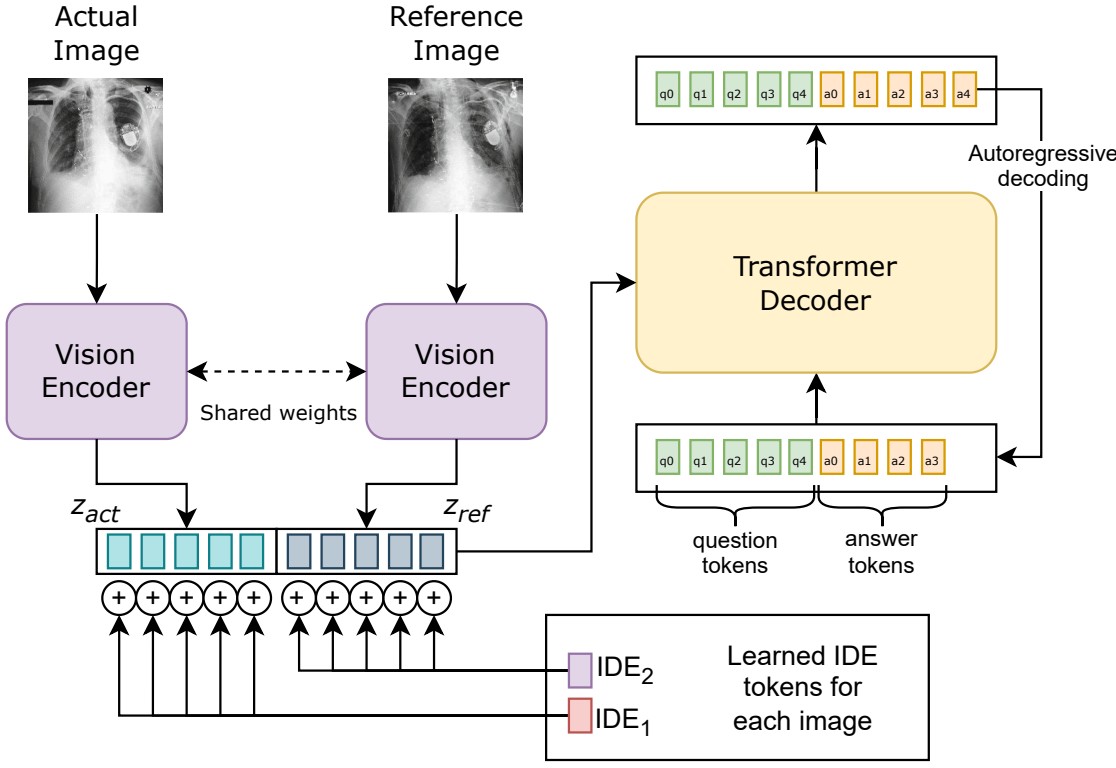

Figure 1: Our proposed architecture for Difference Medical Visual Question Answering.

we learn two tensors of dimension $d$, where $d$ is the hidden size of the decoder. These tensors are learned concurrently with the entire model through backpropagation during training and are then kept fixed during inference.

For each image, one of these tensors is added across all its sequence tokens. Specifically, for the feature tensor of the first image $F \in \mathbb{R}^{t \times d}$, we add the first learned tensor $IDE_1 \in \mathbb{R}^{1 \times d}$ to each of the $t$ tokens of the first image. The same procedure applies to the second image using the second learned tensor $IDE_2$. In Appendix A we show how we implement IDE in PyTorch. This mechanism ensures that each image retains a distinct differentiating signal throughout the processing pipeline, as reflected in the performance improvements shown in Table 1.

### 3.3. Language Component

Our choice of a 3-layer transformer decoder is motivated by several considerations. First, the dataset features a very limited vocabulary of only 101 words, and the questions and answers exhibit a similar structure, reducing the need for a large, heavily pretrained model with an extensive vocabulary. We conducted experiments with decoder configurations of 2, 3, and 4 layers. The 3-layer setup emerged as the optimal configuration—2 layers resulted in inferior performance, and although 4 layers matched the performance of the 3-layer

model, it imposed higher computational costs without additional benefits. Furthermore, our experiments with a pre-trained BERT-base decoder (augmented with new cross-attention layers) yielded significantly worse performance, indicating that a pre-trained decoder might not align well with our task's specific characteristics. Therefore, a lightweight, 3-layer transformer decoder strikes the best balance between computational efficiency and effective fusion of visual and textual information for autoregressive decoding in the medical question-answering context.

The decoder has a hidden size of 1024, an intermediate size of 4096, and employs GELU activation. It comprises both self-attention layers and cross-attention layers. In this design, the cross-attention mechanism leverages the final-stage output of the Swin Transformer as keys and values, while the decoder's self-attention layers process the textual context. This arrangement enables effective fusion of visual and textual information, facilitating accurate autoregressive decoding of medical question answers. Notably, this lightweight decoder, containing approximately 51M parameters, is significantly smaller than the PLURAL model (Cho et al., 2024), which has 184M parameters.

### 3.4. Training Strategy

Our training strategy is a three-stage process aimed at optimizing the model's vision and language components sequentially before integrating them for the VQA task.

In the first stage, we fine-tune the Swin Transformer on the MIMIC-CXR dataset with CheXpert labels. Since the Medical-Diff-VQA dataset is derived from the MIMIC-CXR dataset, we ensure consistency by using the same dataset splits. The AdamW (Loshchilov and Hutter, 2019) optimizer is used with a learning rate of $1 \times 10^{-4}$, a batch size of 24, and a weight decay of 0.05. Training is conducted for 30 epochs using a Cosine Annealing learning rate scheduler. The best-performing model is selected based on the validation loss.

In the second stage, the fine-tuned Swin model is integrated with the transformer decoder to construct the Vision Encoder-Decoder (VED) architecture. During this phase, the parameters of the Swin model remain frozen, allowing the training to focus exclusively on the decoder. The decoder is trained for 20 epochs using a batch size of 64, employing the Adam optimizer (Kingma and Ba, 2017) with a learning rate of $3 \times 10^{-4}$

In the third stage, we unfreeze the Swin model and fine-tune the entire VED architecture for 20 more epochs with the learning rate set to $3 \times 10^{-6}$. Since we are training the entire model, we use a smaller batch size of 8 but simulating a batch size of 64 with gradient accumulation to reduce memory consumption.

The model is trained using the negative log-likelihood loss function, which calculates the probability of the correct answer given the question and the images. During training, the learning rate is linearly decreased, and Hard Negative Mining (Sung and Poggio, 1998) is employed to select the most challenging samples for further training. To improve the quality of generated text, beam search with a beam size of 2 is used during inference, enabling the model to explore multiple possible outputs and select the most likely answer. For image inputs, a series of augmentation techniques—including shift, scale, rotation, and brightness/contrast adjustments—are applied to simulate various real-world conditions and enhance the model's robustness.

### 3.5. Medical-Diff-VQA dataset

The Medical-Diff-VQA (Hu et al., 2023) dataset is a comprehensive resource specifically designed for the Difference VQA task in the medical domain. It focuses on answering questions about differences between pairs of chest X-ray images, reflecting a radiologist's typical workflow of comparative analysis for diagnostic evaluation.

Derived from the MIMIC-CXR (Johnson et al., 2019) database, the dataset comprises over 164,000 pairs of main and reference chest X-ray images, resulting in a total of 700,703 question-answer pairs. These questions are categorized into seven types: abnormality, location, type, level, view, presence, and difference. The "difference" category, which focuses on identifying changes between two images, includes 164,324 question-answer pairs, making it the most relevant subset for this study.

For the purposes of this work, only the "difference" question type is considered, as it aligns with the current state-of-the-art focus in the field and offers the most direct application for advancing medical diagnostic tools. By leveraging this targeted subset, the study aims to enhance model performance in generating precise and clinically relevant answers.

### 3.6. Framework Overview

Our framework integrates a vision encoder with a transformer decoder in a three-stage training process:

- **Input:** Two preprocessed 384×384 chest X-ray images and tokenized question.

- **Vision:** A Swin Transformer extracts hierarchical features, enhanced by an Image Differentiation Embedding (IDE) to distinguish the images.

- **Language:** A lightweight 3-layer transformer decoder fuses visual features with textual context via cross-attention to generate answers.

- **Training:** We fine-tune the vision module, train the decoder with frozen vision parameters, and finally fine-tune the entire model end-to-end.

## 4. Experimentation and Results

All experiments were conducted on a single NVIDIA RTX 4090 GPU with 24GB of memory, ensuring a high level of computational efficiency and support for complex model architectures. Our implementation utilized the PyTorch deep learning framework (version: 2.1.1) in conjunction with the Hugging Face Transformers library (version: 4.27.4), providing a robust and flexible platform for training and fine-tuning our models.

For evaluation, we adhered closely to the metrics and methodologies outlined in the original paper, all token-based. Specifically, we employed BLEU (Papineni et al., 2002), METEOR (Banerjee and Lavie, 2005), ROUGE-L (Lin, 2004), and CIDEr (Vedantam et al., 2015) as performance metrics to assess the effectiveness of our model. Also, we have added BERTScore (Zhang et al., 2020) to have a score closer to a personal judgment, which token-based metrics cannot provide. The improvement in performance through each stage is shown in Table 1 and the comparision of our model with state-of-the-art methods on the Medical-Diff-VQA dataset is shown in Table 2.

Table 1: Comparison of our model performance through all the stages. The best results are shown in **bold**.

| Model | BLEU-1 | BLEU-2 | BLEU-3 | BLEU-4 | METEOR | ROUGE-L | CIDEr | BERTScore |
|---|---|---|---|---|---|---|---|---|
| Baseline | 0.546 | 0.486 | 0.436 | 0.385 | 0.317 | 0.589 | 1.492 | 0.614 |
| + IDE | 0.673 | 0.600 | 0.541 | 0.486 | 0.358 | 0.628 | 1.698 | 0.658 |
| + Train Decoder FT-Swin | 0.703 | 0.632 | 0.572 | 0.516 | 0.385 | 0.659 | 1.849 | **0.706** |
| + Unfreeze Swin | **0.716** | **0.647** | **0.590** | **0.537** | **0.389** | **0.670** | **2.119** | 0.704 |

The results in Table 1 highlight the continuous improvement achieved through our multi-stage training strategy. Starting from a baseline where we trained the entire VED architecture, with the decoder starting from scratch and the Swin model initialized using its pretrained weights on ImageNet-21k, the results reflect basic performance without domain-specific optimization. Incorporating IDE introduces a significant boost in performance across all metrics. In the second stage training the decoder while the fine-tuned Swin remains fronzen results in additional performance gains. Finally, in the third stage unfreezing the Swin model for joint optimization leads to the best results across all evaluation metrics.

Table 2: Comparison of our model with state-of-the-art methods on the Medical-Diff-VQA dataset. The best results are shown in **bold**.

| Methods | BLEU-1 | BLEU-2 | BLEU-3 | BLEU-4 | METEOR | ROUGE-L | CIDEr |
|---|---|---|---|---|---|---|---|
| MCCFormers (Qiu et al., 2021) | 0.214 | 0.190 | 0.170 | 0.153 | 0.319 | 0.340 | 0 |
| IDCPCL (Yao et al., 2022) | 0.614 | 0.541 | 0.474 | 0.414 | 0.303 | 0.582 | 0.703 |
| EKAID (Hu et al., 2023) | 0.628 | 0.553 | 0.491 | 0.434 | 0.339 | 0.557 | 1.027 |
| EIE-all (Wang et al., 2024) | 0.646 | 0.583 | 0.528 | 0.477 | **0.401** | 0.653 | 1.698 |
| RegioMix (Yung et al., 2024) | 0.705 | 0.633 | 0.572 | 0.517 | 0.381 | 0.651 | 1.804 |
| PLURAL (Cho et al., 2024) | 0.704 | 0.633 | 0.575 | 0.520 | 0.381 | 0.653 | 1.832 |
| Ours | **0.716** | **0.647** | **0.590** | **0.537** | 0.389 | **0.670** | **2.119** |

The results presented in Table 2 demonstrate the superior performance of our model compared to state-of-the-art methods on the Medical-Diff-VQA dataset. Our model achieves the highest scores across all BLEU metrics, with 0.716 in BLEU-1 and 0.537 in BLEU-4, highlighting its ability to produce precise and linguistically faithful answers. While the METEOR score (0.389) is slightly lower than EIE-all (0.401), our model achieves the best performance in ROUGE-L (0.670), indicating strong coherence and structural accuracy. Notably, the CIDEr score of 2.119 significantly surpasses all competing methods, emphasizing our model's ability to generate clinically relevant and contextually accurate responses. These results underscore the effectiveness of our approach in addressing the Medical VQA task, setting a new benchmark for both linguistic fidelity and domain-specific relevance.

Despite good results that even outperform the state-of-the-art, more research is needed to improve some current limitations. For example, Figure 2 shows two scenarios where all metrics are close to the highest possible values. In the first scenario, the high metric values reflect the strong alignment between the model's predictions and the ground truth, highlighting its accuracy in this case.

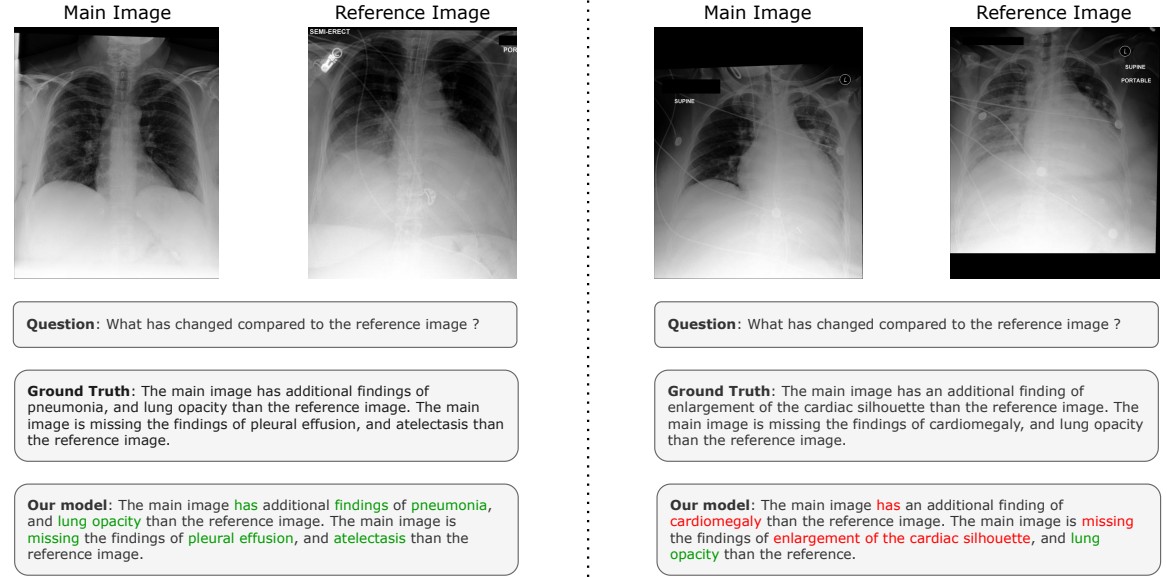

Figure 2: Examples of difference questions and their corresponding answers generated by our model and the ground truth. Correct predictions are highlighted in green, while incorrect predictions are highlighted in red.

The issue arises in the second scenario, where the metrics are 1.00, 0.968, 0.911, and 0.853 for BLEU, 0.589 for METEOR and 0.818 for ROUGE-L, values very close to the maximum and also exceeding those reported in Table 2, yet the model's predicted response exhibits diagnostic errors. Upon closer inspection, it becomes evident that, while the model correctly identifies the anomalies present in the radiograph, it misinterprets the diagnosis, either assuming the resolution of certain anomalies when they have not subsided or identifying new ones that are not actually present. These results may suggest that the model's outputs are of high quality, yet they underscore a critical issue: high performance on these metrics does not necessarily translate to correctness in a clinical diagnostic context.

This observation opens a crucial debate about the appropriateness of existing evaluation metrics for tasks like Medical Visual Question Answering. It emphasizes the need to develop new metrics tailored to this domain, which go beyond assessing the linguistic or structural quality of the generated answers. For tasks involving medical diagnostics, it is essential to evaluate not only the fluency and relevance of the generated response but also its accuracy in identifying and interpreting anomalies. Addressing this gap is vital to ensure that models deployed in clinical settings contribute to accurate and reliable decision-making.

## 5. Conclusion

In conclusion, the evaluation of the proposed Difference Medical VQA model underscores its potential to assist in clinical decision-making through automated and accurate image

analysis. The model demonstrated strong alignment with the ground truth in detecting prominent medical findings, suggesting its reliability in handling straightforward diagnostic tasks. Its structured and contextually relevant responses further highlight its utility in providing actionable insights.

However, the model's sensitivity to nuanced or complex medical differences, such as distinguishing subtle anatomical changes, remains an area for improvement. Addressing these limitations will require enhancing the training process with more diverse and finely annotated datasets, along with implementing advanced mechanisms for error analysis and fine-grained feature detection.

Despite these challenges, the model's superior performance metrics compared to state-of-the-art approaches, particularly in generating clinically relevant and linguistically accurate responses, set a new benchmark for Difference Medical VQA systems. These findings suggest that with continued refinement, this approach holds promise for advancing diagnostic precision and supporting radiological workflows.

## Acknowledgments

This work was supported by the Generalitat Valenciana under the grant CIPROM/2023/17.

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

## Appendix A. IDE Implementation

**Algorithm 1** IDE PyTorch Pseudocode

```
    # DEFINE IDE
    IDE = torch.nn.Parameter(2, decoder_dim) # Learnable IDE tensors

def forward():
    imgs_features = encode(images) # (b, 2, 144, 1024)
    # IMAGES FEATURES ARE (batch_size, num_images, seq_len, decoder_dim)
    IDE = IDE.unsqueeze(1) # (2, 1024) -> (2, 1, 1024)
    imgs_features += IDE
    features_concat = rearrange(imgs_features, 'b d1 d2 d3 -> b (d1 d2) d3')
```

