# OpenReview forum: "Unveiling Differences: A Vision Encoder-Decoder Model for Difference Medical Visual Question Answering"
_MIDL.io/2025/Conference — MIDL 2025 Oral_

### Official Review · Reviewer_dHhS · 2025-02-21

**Confidence:** 5
**Preliminary Rating:** 3
**Final Rating:** 4

**Summary:**

The paper introduces a Vision Encoder-Decoder (VED) model for Difference Medical Visual Question Answering (Diff-VQA), focusing on detecting changes in chest X-rays. It uses a lightweight Transformer decoder and enhanced fusion mechanism.

**Strengths:**

The paper is well organized and the experiment results looks good. The paper uses Swin Transformer and a lightweight BERT decoder, achieving state-of-the-art performance. The ablation study proves each mosule has contribution to the results.

**Weaknesses:**

Explanation to the IDE module seems not clear. It would be better if more details can be provided. In the evaluation section, the paper provides some metrics, howerer, N-grams metrics seems not suitablle for this kind of medical report, some papers has pointed this out. It would be better to add the evaluation results using LLM-based method.

**Detailed Comments:**

1. Why did you choose Swin Transformer as the visual encoder? Compared with other visual encoders (such as ResNet or ViT), what are the unique advantages of Swin Transformer in processing medical images? When choosing the visual encoder, did you consider other pre-trained models (such as ViT or EfficientNet)? If so, why did you finally choose Swin Transformer?
2. Why was the BERT decoder reduced from 12 layers to 3 layers? Does this design strike the best balance between performance and computational efficiency?When reducing the number of decoder layers, were other lightweight architectures (such as TinyBERT or DistilBERT) considered? If so, why was 3-layer BERT ultimately chosen?
3. How is IDE implemented? How is the dimension d of the embedding vector determined? During training, are the learning rates and optimization strategies of IDE the same as other parts (such as the visual encoder and decoder)? If so, what are the differences?
4. In addition, the second sub figure of fig.2, the ground truth text is also highlighted as red, is this a typo?

**Justification Of The Final Rating:**

The responses provided in the rebuttal, along with the revisions in the manuscript, effectively address the reviewers’ questions and concerns. The authors have conducted sufficient experiments to support their claims, and the manuscript is well-structured and clearly presented. Given the thoroughness of the experimental validation and the clarity of the revisions, this paper should be considered for acceptance.

**Justification Of The Preliminary Rating:**

The paper is well organized and the experiment results looks good. However, the explanation of the modules, espeacially of IDE, needs to be more detailed. And it would be better if the author have more explanation to the vision encoder and text encodr selection.

**Questions To Address In The Rebuttal:**

1. Why did you choose Swin Transformer as the visual encoder? Compared with other visual encoders (such as ResNet or ViT), what are the unique advantages of Swin Transformer in processing medical images? When choosing the visual encoder, did you consider other pre-trained models (such as ViT or EfficientNet)? If so, why did you finally choose Swin Transformer?
2. Why was the BERT decoder reduced from 12 layers to 3 layers? Does this design strike the best balance between performance and computational efficiency?When reducing the number of decoder layers, were other lightweight architectures (such as TinyBERT or DistilBERT) considered? If so, why was 3-layer BERT ultimately chosen?
3. How is IDE implemented? How is the dimension d of the embedding vector determined? During training, are the learning rates and optimization strategies of IDE the same as other parts (such as the visual encoder and decoder)? If so, what are the differences?

---

> ### Author Response · Authors · 2025-03-07
> **Official Commento to reviewer dHhS**
>
> The authors sincerely thank reviewers for their thoughtful and constructive feedback. We have carefully revised the manuscript based on these comments, leading to significant improvements in clarity, structure, and content. Below, we address each point in detail.
>
> ---
>
> ### General Changes
>
> **Transformer Decoder:**
>
> We have renamed the “BERT decoder” to “Transformer Decoder” to more accurately reflect its design. Although inspired by BERT, our decoder neither uses pre-trained BERT weights nor follows its training strategy. The decoder is initialized with random weights and trained from scratch. Details on its configuration and justification are now provided in Section 3.3 of the revised manuscript.
>
> Regarding the questions in Fig. 2, while some may appear identical, not all of them are exactly the same. The transformer decoder processes the question as context and generates the answer token by token in an autoregressive manner.
>
> **Image Differentiation Embedding (IDE) Component:**
>
> We have clarified the IDE mechanism’s description in the revised manuscript. IDE involves learning two distinct embedding tensors—each with dimension *d* (matching the decoder’s hidden size)—that are added element-wise to every token of their corresponding image’s feature representation. These tensors are learned alongside the model through backpropagation and remain fixed during inference. This ensures each image retains a unique differentiating signal throughout the fusion process. Additional implementation details are now provided in the Appendix.
>
> ---
>
> ### Reviewer Comments and Responses
>
> **1. Why was Swin Transformer chosen as the visual encoder over alternatives like ResNet, ViT, or EfficientNet?**
>
> We have added the justification for selecting Swin Transformer to Section 3.1 of the revised manuscript. In our experiments, both EfficientNet and TinyViT underperformed, while ViT achieved comparable results to Swin Transformer but with increased model size and reduced computational efficiency. Given our computational constraints and Swin Transformer’s superior performance-to-efficiency ratio in processing medical images, it was the most suitable choice.
>
> **2. Why was the BERT decoder reduced from 12 to 3 layers? Were lightweight alternatives like TinyBERT or DistilBERT considered?**
>
> To clarify, our 3-layer architecture is not a reduction from the standard 12-layer BERT model. As detailed in our general changes, we designed the Transformer Decoder from scratch, inspired by BERT’s architecture but without using pre-trained weights or its training strategy. Empirical evaluation showed that the 3-layer configuration achieves an optimal balance between computational efficiency and performance for our task. Further justification is now provided in Section 3.3.
>
> **3. How is IDE implemented, and how were its parameters determined?**
>
> A more detailed explanation of IDE is now provided in Section 3.2 and illustrated in Fig. 1 of the revised manuscript. IDE helps the model distinguish between current and reference images by learning two separate embedding tensors added to the image feature representations. The embedding dimension *d* is set to match the image feature tensor size (*d* = 1024). IDE functions similarly to learned positional encodings, with distinct tokens per image replicated across all feature vectors. These tokens are optimized alongside the rest of the model parameters during training.
>
> **4. In Fig. 2, why is the ground truth text highlighted in red? Is this a typo?**
>
> The second subfigure in Fig. 2 illustrates a case where the answer achieves a high BLEU score because all words from the ground truth appear in the generated answer and vice versa. Our intent was to emphasize the failure of the answer despite the good BLEU score. Initially, readers found it difficult to identify the issue, so we highlighted the ground truth text for clarity. However, we have reconsidered this approach and removed the highlighting in the revised manuscript.
>
> ---
>
> We once again thank reviewers for their invaluable feedback and hope that the revisions have adequately addressed all concerns.

---

> > ### Comment · Reviewer_dHhS · 2025-03-13
> >
> > Dear Authors,
> >
> > I appreciate your detailed responses and the revisions made to clarify key aspects of your work. Below are comments regarding your responses:
> >
> > 1. Transformer Decoder: The renaming from “BERT decoder” to “Transformer Decoder” is a welcome change as it more accurately reflects the model’s architecture. Your clarification that it does not use pre-trained BERT weights or its training strategy is helpful. The added explanation in Section 3.3 should aid in resolving any confusion.
> > 2. Image Differentiation Embedding (IDE) Component: The additional details on IDE’s learning mechanism and role in the fusion process enhance our understanding. Providing an explanation of how the embedding tensors are learned and used during inference is particularly useful. The inclusion of further implementation details in the Appendix strengthens the technical clarity.
> > 3. IDE Implementation and Parameterization: The additional explanation provided in Section 3.2 and the illustration in Fig. 1 improve clarity. Explicitly defining the embedding dimension and its similarity to learned positional encodings makes the mechanism more interpretable. Your response effectively addresses this concern.
> > 4. Fig. 2 Ground Truth Highlighting: This change ensures that the figure remains informative without unnecessary visual distractions.
> >
> > Overall, your revisions and clarifications enhance the manuscript’s clarity. Thank you for addressing the concerns in a thorough and structured manner.

---

### Official Review · Reviewer_4v8G · 2025-02-28

**Confidence:** 4
**Preliminary Rating:** 3
**Recommendation:** Poster
**Final Rating:** 4

**Summary:**

This work presents a DL-based framework that could compare the difference between two medical images and explain them in natural languages. The main architecture consists of a swim transformer that encodes the image into an embedding, and an LLM that takes the question and image embeddings and translates them to human-interpretable answers.

**Strengths:**

1. The motivation is well delivered;
2. The overall writing is smooth and easy to follow;
3. Experiment results are solid;
4. Further discussions about the failure cases and limitations of existing evaluation methods are instructive.

**Weaknesses:**

1. Manuscript needs restructuring;
2. Writing style is inconsistent;
3. Minor language issues;
4. Lacking methodology overview;
5. Some key elements in the proposal aren't well-defined;
6. More discussions about the experiments are preferred.

**Detailed Comments:**

1. The writing style is inconsistent, especially in Section 2.1. It looks like these paragraphs are from different people.
2. In Section 1, isn't the training strategy you present one of the key contributions of this paper? I only see you mentioning the key contribution being the light-weight transformer text decoder.
3. In Section 1, the paragraph about the dataset doesn't fit here, merge it with Section 2. Also, the structure of Section 2 is inappropriate, you should only talk about related works and move the discussion about the dataset to experiments or implementation.
In section 2, there's no need to start a new paragraph for each work you cite.
4. Do case checking for proper nouns and common nouns, such as "Dual Alignment module", "Light-Weight Transformer Text Decoder".
5. In Section 3, write a framework overview to briefly explain the overall workflow of your proposal.
6. In Section 3.2, I'm confused by this "learnable embedding matrix", what do you mean "learnable", is it trained once and kept fixed during inference regardless of input images, or it is dynamically predicted by something else? Where does it come from?
7. In Section 3.3, I don't understand "the BERT decoder's output is used as the query input", doesn't BERT decoder output the final answer (as depicted in Fig. 1)? Why is it used as the query input?
In Section 4, I notice that the questions in Fig.2 are all the same, why border inputting this to BERT if it doesn't change? Does it make a difference if you change the question? You mention that the answer sometimes misinterprets the diagnosis, what if you change the question to be more specific and to be focusing on only a meaningful ROI? Does the model better capture the diagnosis in its answer?

**Justification Of The Final Rating:**

The revision has successfully addressed most of my concerns, and the new version looks good from the conference's standard. Although I'm not an expert in LLMs, I would recommend accepting if there are no other issues from other reviewers.

**Justification Of The Preliminary Rating:**

This paper presents an interesting method in developing a language model that is capable of describing the semantic difference between two medical images. The model architecture is novel and the experiment results are convincing. But several major weaknesses need to be addressed before acceptance.

**Questions To Address In The Rebuttal:**

1. Writing style;
2. Case checking;
3.Related work;
4. Methodology overview;
5. More experiment discussions.

**Special Issue:**

No

---

> ### Author Response · Authors · 2025-03-07
> **Official Commento to reviewer 4v8G**
>
> The authors sincerely thank reviewers for their thoughtful and constructive feedback. We have carefully revised the manuscript based on these comments, leading to significant improvements in clarity, structure, and content. Below, we address each point in detail.
>
> ---
>
> ### General Changes
>
> - **Manuscript Restructuring and Writing Style:**
>   We have restructured the manuscript to improve clarity and readability. In addition, inconsistencies in writing style and minor language issues have been addressed throughout the paper.
>
> - **Methodology Overview:**
>   A new Section 3.6 titled “Framework Overview” has been added to provide a comprehensive overview of our methodology, addressing concerns about the lack of detailed methodological context and the need for better-defined proposal elements.
>
> - **Proposal Clarification:**
>   In response to feedback regarding some key elements of the proposal being under-defined, we clarify that the training strategy is not a novel key contribution in this paper.
>
> - **Question Refinement:**
>   While refining questions to focus on specific regions of interest might improve diagnostic capture, we maintained the original dataset settings for fair comparison with existing methods. This potential improvement is noted for future work.
>
> - **BERT Decoder:**
>   In our revised manuscript, we have renamed the “BERT decoder” to “Transformer Decoder” to better reflect its design. Although our decoder architecture is inspired by BERT, we do not utilize pre-trained BERT weights or its training strategy. Instead, the decoder is initialized with random weights and trained from scratch. We have detailed the configuration parameters and provided further justification in Section 3.3 of the revised manuscript.
>
> - **IDE Component:**
>   In our revised manuscript, we have clarified the description of the Image Differentiation Embedding (IDE) mechanism to address earlier reviewer concerns regarding its functionality. In the updated version, we explain that IDE involves learning two distinct embedding tensors—each of dimension d, matching the hidden size of the decoder—that are added element-wise to every token of their corresponding image's feature representation. These tensors are learned concurrently with the entire model through backpropagation during training and are then kept fixed during inference. This approach ensures that each image retains a unique, differentiating signal throughout the fusion process. We have also provided a detailed implementation in the Appendix.
>
> ---
>
> ### Reviewer Comments and Responses
>
> **1. Methodology Overview**
> A new Section 3.6 is now added with the title "Framework Overview." Thank you for the recommendation.
>
> **2. In Section 4, I notice that the questions in Fig. 2 are all the same. Why bother inputting this to BERT if it doesn't change? Does it make a difference if you change the question? You mention that the answer sometimes misinterprets the diagnosis—what if you change the question to be more specific and focus only on a meaningful region of interest (ROI)? Does the model better capture the diagnosis in its answer?**
> Thank you for this interesting suggestion. Refining the question to focus on a specific, meaningful ROI could indeed help the model capture the diagnosis more accurately. However, in our current study, we chose not to extensively modify or preprocess the dataset to ensure that the results remain comparable with other state-of-the-art methods. We recognize the potential benefits of this approach and look forward to exploring it in future work.
>
> **3. In Section 3.2, I'm confused by this "learnable embedding matrix"—what do you mean by "learnable"? Is it trained once and kept fixed during inference regardless of input images, or is it dynamically predicted by something else? Where does it come from?**
> IDE is now explained in Section 3.2 in a clearer way. IDE is a matrix or tensor that is learned during training through backpropagation and then kept fixed during inference. This approach ensures the model captures distinct signals for each image while maintaining computational efficiency.
>
> ---
>
> We once again thank reviewers for their valuable insights and suggestions. We believe the revisions have addressed all concerns and improved the overall quality of the manuscript.

---

> > ### Comment · Reviewer_4v8G · 2025-03-13
> >
> > Thanks the authors for their job in the revision. The update has much improved compared to the previous one. I have no more concerns.

---

### Official Review · Reviewer_PR54 · 2025-02-28

**Confidence:** 5
**Preliminary Rating:** 4
**Recommendation:** Poster
**Final Rating:** 5

**Summary:**

This paper introduces a Vision Encoder-Decoder architecture for Difference Medical Visual Question Answering (VQA), specifically comparing chest X-ray images to detect changes. The key ideas involve a lightweight Transformer decoder and an Image Differentiation Embedding mechanism to enhance image comparison. Experiments consisted of a three-stage training process and evaluation on the Medical-Diff-VQA dataset, demonstrating state-of-the-art performance in standard VQA metrics.

**Strengths:**

1. The model employs a novel three-stage training methodology, demonstrating progressive performance gains throughout the ablation study.
2. The model achieves competitive scores on the chosen evaluation metrics, surpassing existing state-of-the-art methods.
3. The authors provide a comprehensive review of prior work and effectively benchmark their model against previous models using the same dataset, strengthening the study's contribution.

**Weaknesses:**

1. The authors' evaluation relies solely on token-based metrics, which, as demonstrated in prior research (e.g., https://aclanthology.org/J18-3002/), are known to be brittle. Employing more robust metrics, such as BERTScore (https://openreview.net/pdf?id=SkeHuCVFDr) or DocLens (https://aclanthology.org/2024.acl-long.39/), which exhibit stronger correlation with human judgment, would have yielded a more reliable assessment.
2. Even in the event that it was not possible to run these metrics a human evaluation study over a subset of the testset would be beneficial.
3. The study lacks statistical significance testing to validate the proposed models' performance against existing state-of-the-art methods.

**Detailed Comments:**

1. Providing a more thorough exposition of the evaluation metrics and their contextual relevance to the task would be advantageous for the reader.

**Justification Of The Final Rating:**

The authors' inclusion of BERTScore significantly strengthens the evaluation and addresses a key weakness. The revised manuscript demonstrates clear improvements and sets a positive precedent for future work in Medical VQA by incorporating more robust, human-aligned evaluation metrics.

**Justification Of The Preliminary Rating:**

The paper shows promise with its novel methodology and thorough benchmarking. However, the fundamental flaws in its evaluation methodology and the lack of statistical rigor prevent it from being a strong accept.

**Questions To Address In The Rebuttal:**

1. Employing more holistic evaluation metrics, which demonstrate stronger correlation with human judgment, would significantly improve assessment compared to traditional metrics.
2. Furthermore, statistical significance testing should be conducted to validate the proposed methods' superiority over existing state-of-the-art approaches.

---

> ### Author Response · Authors · 2025-03-07
> **Official Commento to reviewer PR54**
>
> The authors sincerely thank reviewers for their thoughtful and constructive feedback. We have carefully revised the manuscript based on these comments, leading to significant improvements in clarity, structure, and content. Below, we address each point in detail.
>
> ---
>
> ### General Changes
>
> **Transformer Decoder:**
>
> We have renamed the “BERT decoder” to “Transformer Decoder” to more accurately reflect its design. Although inspired by BERT, our decoder neither uses pre-trained BERT weights nor follows its training strategy. The decoder is initialized with random weights and trained from scratch. Details on its configuration and justification are now provided in Section 3.3 of the revised manuscript.
>
> Regarding the questions in Fig. 2, while some may appear identical, not all of them are exactly the same. The transformer decoder processes the question as context and generates the answer token by token in an autoregressive manner.
>
> **Image Differentiation Embedding (IDE) Component:**
>
> We have clarified the IDE mechanism’s description in the revised manuscript. IDE involves learning two distinct embedding tensors—each with dimension *d* (matching the decoder’s hidden size)—that are added element-wise to every token of their corresponding image’s feature representation. These tensors are learned alongside the model through backpropagation and remain fixed during inference. This ensures each image retains a unique differentiating signal throughout the fusion process. Additional implementation details are now provided in the Appendix.
>
> ---
>
> ### Reviewer Comments and Responses
>
> **1. Employing more holistic evaluation metrics, which demonstrate stronger correlation with human judgment, would significantly improve assessment compared to traditional metrics.**
>
> In our experiments, we indeed computed metrics such as BERTScore, which are known to better correlate with human judgment. However, we opted not to report these results in the original paper because the current state-of-the-art methods in Medical VQA do not provide values for these metrics, thereby precluding a direct comparison. Nevertheless, we agree that incorporating these holistic evaluation metrics can offer valuable insights. In the revised version of the manuscript, we have added the results obtained from BERTScore. We believe that including these results not only strengthens our evaluation but also sets a precedent for future work in Medical VQA to adopt more human-aligned assessment measures.
>
> **2. Statistical significance testing should be conducted to validate the proposed methods' superiority over existing state-of-the-art approaches.**
>
> We certainly recognize the value of incorporating rigorous statistical significance testing to further validate our model’s improvements. In an ideal scenario, we would have conducted tests such as paired t-tests or bootstrapping to confirm that our gains are not due to random chance. However, the large scale of our experiments combined with the resource-intensive nature of such tests posed significant challenges. It’s also worth noting that several state-of-the-art studies in the Medical VQA domain, including works like RegioMix and EIE, have similarly opted not to include statistical significance testing.
>
> ---
>
> We once again thank reviewers for their invaluable feedback and hope that the revisions have adequately addressed all concerns.

---

> > ### Comment · Reviewer_PR54 · 2025-03-10
> > **Response to Author Rebuttal**
> >
> > Dear Authors,
> >
> > Thank you for your detailed rebuttal and the revisions made to the manuscript. I appreciate the effort to address the concerns raised in the review. Here are my thoughts on your responses:
> >
> > **Transformer Decoder and IDE Clarification:**
> > 1. Renaming "BERT Decoder" to "Transformer Decoder" is a good step towards clarity.
> > 2. The additional details provided for the IDE mechanism in the appendix are helpful.
> >
> > **Holistic Evaluation Metrics (BERTScore):**
> > 1. I am pleased to see that you have incorporated BERTScore results. This addresses a significant weakness in the original evaluation. While I understand the challenge of comparing with existing methods that may not report these metrics, including them sets a positive precedent.
> >
> > **Statistical Significance Testing:**
> > 1. While I understand the resource constraints, the lack of statistical significance testing remains a concern. Future works in the field should strive to include it as a metric.

---

### Comment · Reviewer_MaP6 · 2025-02-25
**Will submit reviews by Feb 26**

Will submit reviews by Feb 26

---

> ### Comment · Area_Chair_1JDU · 2025-02-27
> **Reviews**
>
> Hi
> Can you submit the reviews at the earliest?

---

### Author Rebuttal · Authors · 2025-03-07

**Rebuttal:**

The ZIP file contains the PDF of the revised manuscript. Changes are highlighted in blue and deleted text is crossed out. Figures 1 and 2 have been updated based on the reviewers' recommendations.

**Supporting Material:**

/attachment/2ad52410075eed174b7f166dfd33a8f769fbab10.zip

---

### Meta-Review · Area_Chair_1JDU · 2025-03-23

**Recommendation:** Accept (Poster)
**Confidence:** 5

**Metareview:**

The paper introduces a Vision Encoder-Decoder model for Difference Medical Visual Question Answering (Diff-VQA) to detect changes in chest X-ray images, leveraging a Swin Transformer-based encoder, a lightweight Transformer decoder, and an Image Differentiation Embedding (IDE) mechanism. Strengths include a well-motivated problem statement, solid experimental results demonstrating state-of-the-art performance, and a comprehensive ablation study validating the proposed methodology. However, key weaknesses involve reliance on token-based evaluation metrics, lack of statistical significance testing, inconsistent writing structure, and insufficient explanation of certain modules, particularly IDE. The authors addressed these concerns in the revision by incorporating stronger evaluation metrics (e.g., BERTScore), improving methodological clarity, and refining the manuscript’s organization. Given the novel contributions, thorough experimental validation, and improvements made during the rebuttal, I recommend acceptance as a poster presentation.